# Low-Temperature Oxidation of Heavy Oil Asphaltene with and without Catalyst

**DOI:** 10.3390/molecules27207075

**Published:** 2022-10-20

**Authors:** Haiyang Yang, Huiyu Yang, Xuemin Yan

**Affiliations:** 1School of Chemistry and Materials Science, Hubei Engineering University, Xiaogan 432000, China; 2College of Chemistry and Environmental Engineering, Yangtze University, Jingzhou 434023, China

**Keywords:** asphaltene, low-temperature catalytic oxidation, hydrogen peroxide, propionic anhydride, methanol

## Abstract

In this study, the asphaltene extracted from Luntai heavy oil was oxidized by a mixture of propionic anhydride and hydrogen peroxide without and with a catalyst. Elemental analysis and infrared spectroscopy results indicated the occurrence of oxygen addition, condensation, and side chain cleavage reactions in the oxidation process. Oxidation products were divided into methanol solubles and methanol insolubles. The H/C and O/C atomic ratios of the MeOHS in the oxidation products without a catalyst were higher than those of the Luntai asphaltene. MeOHS had fewer aromatic rings than Luntai asphaltene. Compared with the oxidative reaction without a catalyst, the total mass of oxidation products and the proportion of MeOHS in oxidation products both increased after catalytic oxidation. This low-temperature oxidation technology can be used to upgrade asphaltenes, and thus can promote the exploitation and processing of heavy oil.

## 1. Introduction

The rapid rise in the world economy is driving a continuous increase in oil demand. Petroleum will remain one of the main energy sources for a long time; however, its production systems are facing very serious challenges. Conventional oil resources cannot meet the demand to support global economic development. Heavy oil has been gradually developed as an unconventional oil resource for its enormous reserves. However, it exhibits high viscosity, large specific gravity, and considerable flow resistance. The viscosity of heavy oil increases exponentially with the increasing asphaltenes’ concentration [1]. The dense flocculations or sediments formed by asphaltenes cause major problems in heavy oil recovery, transportation, and refining [2,3], thereby obstructing oil pipelines [4]. Low-temperature catalytic oxidation is a promising, low-cost, effective, and environmentally friendly technology for heavy oil upgrading. However, upgrading asphaltenes of heavy oil by low-temperature catalytic oxidation is difficult since asphaltene is the most complex component in heavy oil [5,6]. Therefore, only a small amount of literature is available describing the low-temperature oxidation of asphaltenes.

Asphaltenes can be oxidized by hydrogen peroxide, air, sodium hypochlorite, or other oxidants. The ruthenium ion catalytic oxidation (RICO) method has a high selectivity for aromatic carbon, and it can decompose asphaltenes into various compounds, including benzoic acid, fatty acid, and carbon dioxide [7]. However, this method is not widely used due to the high cost of catalysts. Zhang [8] extracted asphaltene from Xinjiang heavy oil and carried out the catalytic oxidation reaction at 150–200 °C. When the temperature was above 180 °C, the cracking reaction rate increased; however, the coke generation rate also increased. Vakhin et al. [9] found that the asphaltene content of heavy oil residue from the Elkhovsky oil refinery increased after catalytic oxidation at 250 °C by air with manganese dioxide (80 wt.%), iron dioxide, and silicon oxide. Nassar et al. [10,11,12,13,14,15,16] adsorbed asphaltenes on nano-catalysts (e.g., CaO, Co_3_O_4_, Fe_3_O_4_, MaO, NiO, and PdO), and then their catalytic oxidation effect was investigated by thermogravimetric analysis at temperatures above 200 °C. The catalytic activities were closely related to the adsorption affinity, acidity and basicity, and morphology of the catalyst surface. The catalytic oxidation process upgraded the asphaltenes, thereby promoting the exploitation and processing of heavy oil. Can asphaltenes be oxidized at low-temperatures below 100 °C?

In this study, hydrogen peroxide and propionic anhydride [17] were used to oxidize Luntai (China) heavy oil asphaltene at low temperatures with and without a catalyst.

## 2. Materials and Methods

### 2.1. Chemicals and Reagents

The heptane-insoluble fraction (asphaltene) was obtained from Luntai heavy oil. The mass concentration of hydrogen peroxide was 30 wt.%, and other reagents such as xylene, n-heptane, propionic anhydride, and methanol were all of analytical grades.

### 2.2. Preparation and Characterization of Catalyst

Iron nitrate nonahydrate and 2-methylimidazole were separately weighed at a 1:8 molar ratio, dissolved in methanol, and then mixed to obtain a homogenous solution. The mixture was placed in a water bath at 30 °C for 12 h, and then left to stand at 50 °C for another 12 h. The solid product was centrifuged, washed with methanol, dried at room temperature, and ground to a fine powder. The powder was calcined under an N_2_ atmosphere at 400 °C at a heating rate of 2.5 °C/min, and then kept at a constant temperature for 2 h. Then the temperature was increased to 1000 °C at a heating rate of 2.5 °C/min, and maintained for 3 h. The black powder obtained after cooling was a carbon-doped nitrogen-supported iron catalyst [18].

The catalyst was characterized by X-ray diffraction (XRD) and scanning electron microscopy (SEM).

### 2.3. Oxidation Experimental Methods

The Luntai asphaltene was ground to a fine powder. About 0.5 g of asphaltene powder, 10 wt.% catalyst relative to asphaltene, and about 6 g propionic anhydride were added to a three-necked flask. The obtained mixture was magnetically stirred in a water bath for 15 min. Then, about 30 mL 30 wt.% hydrogen peroxide was added, and the mixture was heated to 50 °C under the hydraulic reflux and reacted for 24 h. The obtained product was dried in an oven at 60 °C. The dried residue was mixed with methanol and then dispersed by ultrasonic vibration, followed by centrifugation to separate methanol solubles (MeOHS) and methanol insolubles (MeOHI). Finally, MeOHS and MeOHI were dried at 60 °C to remove methanol. The effect of oxidation was evaluated by the proportion of MeOHS in the oxidation product.

### 2.4. Analysis Methods

The elemental compositions of Luntai asphaltene, MeOHS, and MeOHI after oxidation of asphaltene were analyzed by a fully automatic VARIO EL Ⅲ instrument.

Infrared spectra of Luntai asphaltene, MeOHS, and MeOHI were obtained by Fourier transform infrared spectrophotometer (Nicolet 6700).

## 3. Results and Discussion

### 3.1. Oxidation of Luntai Asphaltene

The Luntai asphaltene powder was oxidated by propionic anhydride and hydrogen peroxide without a catalyst. The oxidation products were separated into MeOHS and MeOHI. Table 1 shows the dosage of reagents and the amount of product generated in the asphaltene oxidation experiment.

It can be seen from Table 1 that the total mass of asphaltene after oxidation is 0.7951 g with an increase of 56 wt.% compared with 0.5105 g of Luntai asphaltene. The proportion of MeOHS was around 44 wt.% of the oxidation products.

The elemental compositions and atomic mass of Luntai asphaltene, MeOHS, and MeOHI after reaction are given in Table 2.

It can be seen from Table 2 that the S content in the total atomic mass decreased after oxidation, indicating that the reaction removed some of the S. The H/C atomic ratio of the MeOHS was larger than that of the Luntai asphaltene, suggesting that the MeOHS contained fatty acids generated by the cleavage of alkyl side chains and oxidation products with a small number of aromatic rings [19]. In contrast, the H/C atomic ratio of the MeOHI was smaller than that of the Luntai asphaltene, indicating that the MeOHI contained more condensation reaction products. The O/C atomic ratios of the MeOHS and MeOHI were both larger than that of the Luntai asphaltene, indicating the occurrence of oxidation reactions. The O/C atomic ratio of the MeOHS was about four times that of the MeOHI, revealing that the proportion of oxygen-containing groups was high and they were easily dissolved in methanol. In addition to the significant increase in the total mass of O atoms, the total mass of C and H atoms increased slightly, and the H/C atomic ratio increased. It is attributed to the addition of a small amount of propionic anhydride, which is combined with asphaltene molecules. The resulting products were not evaporated with the solvent during the drying process and remained in the oil sample.

Infrared spectra of Luntai asphaltene, MeOHS, and MeOHI after oxidation reaction are shown in Figure 1.

Comparing the infrared spectra before and after the oxidation reaction, it can be seen that the peak near 1711 cm^−1^ was significantly enhanced, assigned to the carboxyl group (C=O) characteristic absorption peak [20]. The peak near 1213 cm^−1^ was also significantly enhanced, corresponding to phenolic hydroxyl or tertiary alcohol. The peak for the C-O bending vibration of primary alcohol appeared near 1037 cm^−1^. These enhanced IR peaks indicated the occurrence of the oxidation reaction, and the resulting oxygen-containing groups included carboxyl, phenolic hydroxyl, tertiary alcohol, and primary alcohol groups. Luntai asphaltene and MeOHI show a strong peak near 750 cm^−1^, corresponding to aromatic rings, while in MeOHS, no strong peak appeared in this region, suggesting a lower number of aromatic rings in MeOHS, compared to Luntai asphaltene and MeOHI.

### 3.2. Catalytic Oxidation of Luntai Asphaltene

#### 3.2.1. Characterization of Catalyst

The catalyst was prepared according to the method described in Section 2. The SEM image and the XRD pattern of the catalyst are shown in Figure 2. The catalyst is in the form of a crystallographic structure of uniform size [21] A rhombic dodecahedron structure with a relatively smooth surface is observed and the particle size of a single catalyst crystal is around 600–800 nm. Meanwhile, it was also found that there were interconnections between the catalyst single crystal, which may be attributed to the lattice cracking induced by high temperature calcination. This crystal structure change can improve the carrier transfer efficiency during the catalytic process, thereby enhancing the catalytic activity [22]. The XRD pattern of the catalyst depicts obvious diffraction peaks at 44°, 65°, and 82° in 2θ, which correspond to elemental iron [23].

The catalytic activity will be judged according to the following two points: the increase in total mass of the asphaltene and the proportion of MeOHS in the oxidation product.

#### 3.2.2. Catalytic Oxidation of Asphaltene

The Luntai asphaltene powder was oxidated by propionic anhydride and hydrogen peroxide with the catalyst prepared in Section 2.2. The oxidation products were separated into MeOHS and MeOHI. Table 3 shows the dosage of reagents and the amount of product generated by asphaltene catalytic oxidation.

It can be seen from Table 3 that the total mass of asphaltene after oxidation is 0.8406 g with an increase of 68 wt.% compared with 0.5018 g of Luntai asphaltene. It indicated that there were many oxygen-containing groups in the oxidative product, and the proportion of MeOHS after the catalytic oxidation was 46 wt.% of the product. Compared with the reaction without a catalyst, the increase in total mass and the proportion of MeOHS both increased after catalytic oxidation.

Infrared spectra of MeOHS and MeOHI after catalytic oxidation reaction are shown in Figure 3.

Infrared analysis results show the occurrence of oxidation reactions, and the resulting oxygen-containing groups include carboxyl, phenolic hydroxyl, tertiary alcohol, and primary alcohol groups. MeOHI shows a strong peak near 750 cm^−1^, corresponding to aromatic rings, while that peak is absent in MeOHS, indicating that MeOHS did not have as many aromatic rings as MeOHI. Therefore, the ferric ions in the catalyst reduce the side chain of the asphaltene molecular structure and the hydrogen on the aromatic ring during the catalytic oxidation process, thereby oxidizing into oxygen-containing compounds such as carbonyl and alcohol hydroxyl groups [24].

Many oxygen-containing groups were added to the product after catalytic oxidation. Compared with the case without a catalyst, the total mass and the proportion of MeOHS both increased after catalytic oxidation. It indicates that the nitrogen-doped, carbon-supported iron catalyst promotes the oxidation reaction of Luntai asphaltene.

## 4. Conclusions

The Luntai asphaltene was oxidized by hydrogen peroxide and propionic anhydride at a low temperature of 50 °C. The total mass of asphaltene after oxidation increased by 56 wt.%. The oxidation products were divided into 44 wt.% MeOHS and 56 wt.% MeOHI. The H/C and O/C atomic ratios of the MeOHS were higher than those of the Luntai asphaltene. MeOHS had fewer aromatic rings than Luntai asphaltene and MeOHI, and the H/C atomic ratio in MeOHS was as large as 1.5.

The Luntai asphaltene was catalytically oxidized by hydrogen peroxide and propionic anhydride at 50 °C with the nitrogen-doped, carbon-supported iron catalyst. The total mass of asphaltene increased by 68 wt.% after catalytic oxidation. The oxidation products were divided into 46 wt.% MeOHS and 54 wt.% MeOHI. Compared with the reaction without a catalyst, the total mass of oxidation products and the proportion of MeOHS both increased after catalytic oxidation.

This low-temperature catalytic oxidation technology can be used to upgrade asphaltenes and promote the exploitation and processing of heavy oil.

## Figures and Tables

**Figure 1 molecules-27-07075-f001:**
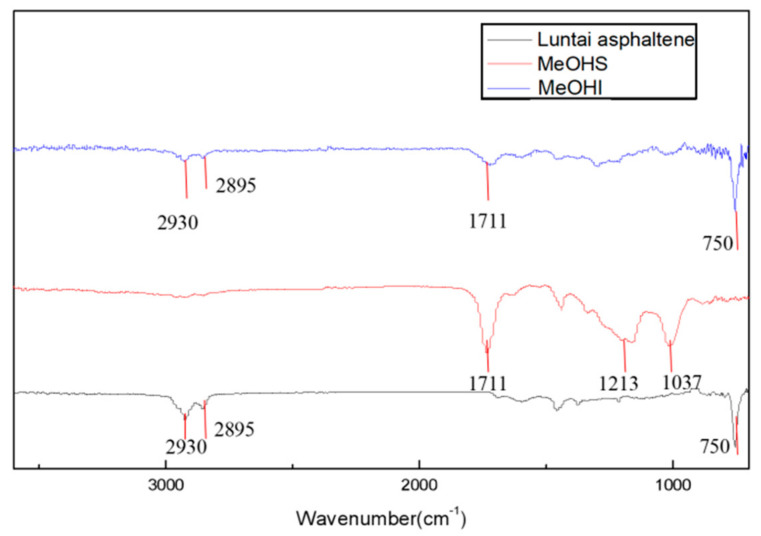
The infrared spectra of Luntai asphaltene, MeOHS, and MeOHI after oxidation reaction: the black line indicates Luntai asphaltene, the red line represents MeOHS, and the blue line shows MeOHI after the oxidation reaction.

**Figure 2 molecules-27-07075-f002:**
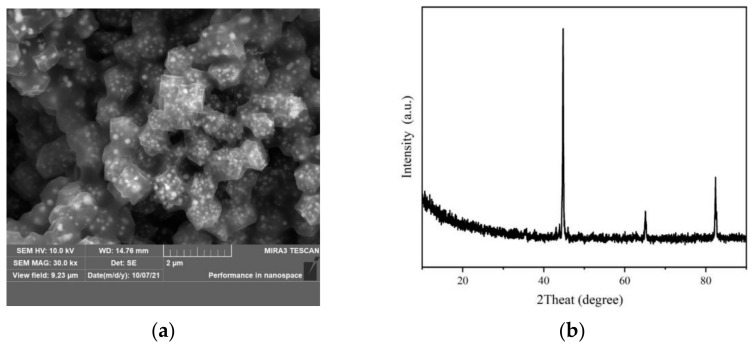
SEM and XRD of the catalyst. (**a**) SEM; (**b**) XRD.

**Figure 3 molecules-27-07075-f003:**
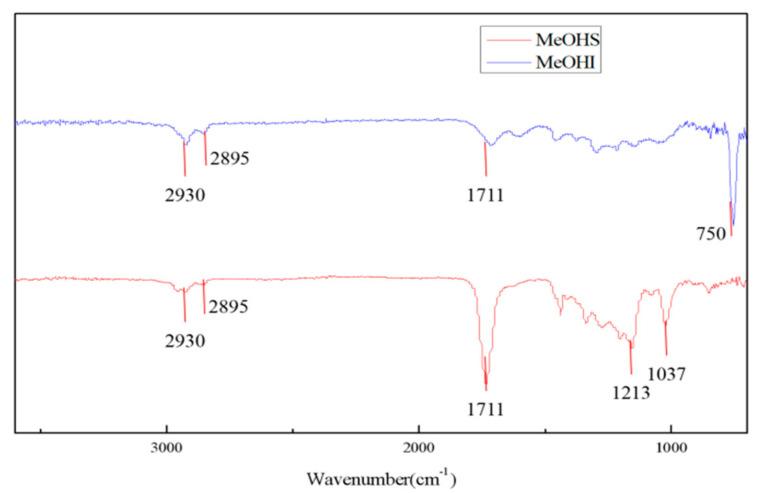
The infrared spectra of MeOHS and MeOHI after catalytic oxidation reaction: the red line represents MeOHS and the blue line indicates MeOHI after the catalytic oxidation reaction.

**Table 1 molecules-27-07075-t001:** Reagent dosage in hydrogen peroxide–propionic anhydride oxidation of asphaltene.

	Luntai Asphaltene	30 wt.% Hydrogen Peroxide	Propionic Anhydride	MeOHS	MeOHI
Amount used or generated	0.5105 g	30 mL	6.2 g	0.3463 g	0.4488 g

**Table 2 molecules-27-07075-t002:** Elemental composition and atomic mass before and after oxidation reaction.

	Elemental Composition, wt.%					Atomic Ratio	
	C	H	S	N	O	H/C	O/C
before oxidation							
Luntai asphaltene, wt.%	84.70	6.11	6.33	1.32	1.54	0.87	0.014
total atomic mass, g	0.4375	0.0316	0.0327	0.0068	0.0079	—	—
total atomic moles, mol	0.0365	0.0316	0.0010	0.0005	0.0005	—	—
after oxidation							
MeOHS, wt.%	45.16	5.60	1.83	0.65	46.76	1.5	0.78
MeOHI, wt.%	71.13	4.70	5.12	1.05	18.00	0.79	0.19
total atomic mass, g	0.4756	0.0405	0.0293	0.0070	0.2427	—	—
total atomic moles, mol	0.0396	0.0405	0.0009	0.0005	0.0152	1.0	0.38
The amount of change in atomic mass before and after oxidation, g	0.0381	0.0089	−0.0034	0.0001	0.2348	—	—

**Table 3 molecules-27-07075-t003:** Reagent dosage in hydrogen peroxide–propionic anhydride catalytic oxidation of asphaltene.

	Luntai Asphaltene	30 wt.% Hydrogen Peroxide	Propionic Anhydride	Catalyst	MeOHS	MeOHI
Amount used or generated	0.5018 g	30 mL	6.1 g	0.0589 g	0.3899 g	0.4507 g

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
