# Peer review of "Low-Temperature Oxidation of Heavy Oil Asphaltene with and without Catalyst"

_molecules, 2022, doi:10.3390/molecules27207075_

Round 1

Reviewer 1 Report (Previous Reviewer 1)

There still exits some mistakes in the paper, the authors should throughly checked.

1)     From the results of the XRD, there only presented diffractive peaks of Fe element, so I wonder why the authors could get the conclusion “nitrogen-doped carbon catalyst successfully supported iron”. Because, without the support, it still showed the same XRD patterns as well.

2)     According to the synthetic process, the MeOHS indicated the obtained products could dissolve in methanol, so I wonder the disappeared peak centered at 750 cm-1 is caused by the substrates contained aromatic rings is insoluble in methanol?

Author Response

Point 1: From the results of the XRD, there only presented diffractive peaks of Fe element, so I wonder why the authors could get the conclusion “nitrogen-doped carbon catalyst successfully supported iron”. Because, without the support, it still showed the same XRD patterns as well.

Response 1: Thank you for your reminder. According to SEM observation, the MOF structure is indeed generated and the particle size is large. There are iron aggregates inevitably, thus affecting the peak shape of XRD. The catalyst particle size is large for the convenience of engineering recycling.

For the convenience of readers, SEM and XRD images are placed together in the new Figure 2.

Point 2: According to the synthetic process, the MeOHS indicated the obtained products could dissolve in methanol, so I wonder the disappeared peak centered at 750 cm-1 is caused by the substrates contained aromatic rings is insoluble in methanol?

Response 2: Thank you for your reminder. According to the different H/C atomic ratios of MeOHS and MeOHI, We suspect that MeOHS are the light components after oxidation. The molecular structures of MeOHS maybe contain more oxygen-containing groups and relatively fewer aromatic rings. However, due to limited conditions, we did not carry out molecular weight analysis of oxidized MeOHS and MeOHI separately. We will analyze the molecular structure of the oxidation products in the subsequent study.

Reviewer 2 Report (New Reviewer)

Review of Manuscript ID: molecules-1936406: “Low-Temperature Oxidation of Heavy Oil Asphaltene With and Without Catalyst” submitted for publication in Molecules journal.

The impression of this manuscript is positive because the article presents interesting and original results, important for both academic and practical audience. which are relatively well explained. I have only two comments shown below. After their addressing I think that this communication can be published in Molecules journal.

1.      On page 4 the statement “Elemental analysis results suggested the H/C atomic ratio in MeOHS was as large as 1.5, indicating that MeOHS can be regarded as a modified light component.” Is difficult to prove, because the hydrogen content in MeOHS (H=5.6 wt.%) is lower than that in the original asphaltenes (H=6.11wt.%). Thus, it is difficult to say that the formed MeOHS is modified light component.

2.      Conclusions section. The conclusion: “Therefore, MeOHS can be regarded as a mod-ified light component.” Needs to be revised addressing comment 1 given above.

Author Response

Point 1: On page 4 the statement “Elemental analysis results suggested the H/C atomic ratio in MeOHS was as large as 1.5, indicating that MeOHS can be regarded as a modified light component.” Is difficult to prove, because the hydrogen content in MeOHS (H=5.6 wt.%) is lower than that in the original asphaltenes (H=6.11wt.%). Thus, it is difficult to say that the formed MeOHS is modified light component.

Response 1: Thank you for your reminder. We have removed the relevant statement. We will use other analytical tools to test whether MeOHS can be regarded as a modified light component in subsequent studies.

Point 2: Conclusions section. The conclusion: “Therefore, MeOHS can be regarded as a mod-ified light component.” Needs to be revised addressing comment 1 given above.

Response 2: Thank you for your reminder. We have made corrections in the Abstrat, Results and Discussion, and Conclusions sections of the manuscript.

This manuscript is a resubmission of an earlier submission. The following is a list of the peer review reports and author responses from that submission.

Round 1

Reviewer 1 Report

1)    In this paper, the authors had investigated the low-temperate oxidation of asphaltene with and with out catalysts, then I suggested the Title of the paper should be modified.

2)    The repeated introduction the oxidation experiments in the Results and Discussion part is redundant.

3)    The discussion of the results should add the relevant references.

4)    I wonder why the XRD results can validate the formation of the N-doped carbon-supported iron catalyst.

5)    The SEM analysis should be more detailed.

6)    Please add the analysis of the function of the catalysts. Furthermore, the activity in this paper should compare with other reported results.

7)    There some grammatical mistakes in the manuscript.

Author Response

Point 1: In this paper, the authors had investigated the low-temperate oxidation of asphaltene with and with out catalysts, then I suggested the Title of the paper should be modified.

Response 1: The focus of the paper is on catalytic oxidation, so the title may remain the same.

Point 2: The repeated introduction the oxidation experiments in the Results and Discussion part is redundant.

Response 2: The repeated introduction the oxidation experiments in the Results and Discussion part was already simplified.

Point 3: The discussion of the results should add the relevant references.

Response 3: Related articles have been added.

Point 4: I wonder why the XRD results can validate the formation of the N-doped carbon-supported iron catalyst.

Response 4: XRD only proves the presence of iron. Modifications have been made in the paper.

Point 5: The SEM analysis should be more detailed.

Response 5: The SEM analysis part in the paper has been revised.

Point 6: Please add the analysis of the function of the catalysts. Furthermore, the activity in this paper should compare with other reported results.

Response 6: There are too few papers on asphaltene oxidation. It's hard to find similar papers to compare the activity of the catalysts. Later studies on catalyst optimization will be carried out.

Point 7: There some grammatical mistakes in the manuscript.

Response 7: The paper has been carefully reviewed and grammatically revised.

Thank you!

Reviewer 2 Report

Please label the IR spectra given in Figures 2 and 4.

Author Response

Response : The figures have been  revised in the paper. 

Thank you!

Reviewer 3 Report

1.      Why the authors used such very high amount of propionic anhydride and hydrogen peroxide (comparing to the amount of asphaltene)?

2.      Can the authors perform the mass balance of the asphaltene?  i.e. can the authors confirm that all asphaltenes are only in MeOHS and MeOHI?

3.      It would be useful for the audience if the author can provide the reaction equation for asphaltene oxidation.

4.      Catalysts were characterized by SEM and XRD.  How this characterization results related to the reaction?

5.      Can the authors characterize the products from the reaction with catalysts (as from non-catalytic reaction)?

Author Response

Point 1: Why the authors used such very high amount of propionic anhydride and hydrogen peroxide (comparing to the amount of asphaltene)?

Response 1: It was difficult to upgrade asphaltenes of heavy oil by low-temperature catalytic oxidation. Sufficient oxidant was used to fully oxidize the asphaltenes. Later studies will investigate the process conditions such as temperature, reaction time, amount of oxidant, and amount of catalyst.

Point 2: Can the authors perform the mass balance of the asphaltene?  i.e. can the authors confirm that all asphaltenes are only in MeOHS and MeOHI?

Response 2: The mass of each element was analyzed and compared before and after the reaction, and only the oxidation element changed greatly before and after the reaction. There were a small amount of light components which might evaporate during the drying stage in the oven. However, they were too few to affect the analysis of the results. So the asphaltenes after oxidation were mainly in MeOHS and MeOHI.

Point 3: It would be useful for the audience if the author can provide the reaction equation for asphaltene oxidation.

Response 3: Asphaltenes are complex mixtures with large molecular weights and complex molecular structures. Obtaining reliable molecular structures of asphaltenes need a lot of effort with advanced analytical methods. Next, we will delve into the molecular structures before and after asphaltene oxidation reactions. Then reliable reaction equations for asphaltene oxidation will be provided.

Point 4: Catalysts were characterized by SEM and XRD.  How this characterization results related to the reaction?

Response 4: SEM and XRD are used to characterize the basic properties of the catalyst. Subsequent studies could optimize catalysts based on these properties.

Point 5: Can the authors characterize the products from the reaction with catalysts (as from non-catalytic reaction)?

Response 5: Infrared analysis and mass analysis are sufficient to illustrate the effect of catalytic oxidation. Elemental analysis cannot tell the difference between the presence and absence of catalyst and is therefore not necessary. Subsequent studies will further study the changes in the molecular structure of asphaltenes after oxidation and catalytic oxidation.

Thank you!

Round 2

Reviewer 1 Report

I think the suggestions or questions offered previously is not carefully solved. Just like the label in Figure 1 is still wrong, MrOHI?

Reviewer 3 Report

Thank you very much for your quick response.  I would like to have this research published in this journal but in a regular article (not in communication).  The authors still have related works in progress as mentioned in the response.  I am looking forward to reading the full-length article.